# India nudges to contain COVID-19 pandemic: A reactive public policy analysis using machine-learning based topic modelling

**Ramit Debnath**[1,2], **Ronita Bardhan**[1]*

**1** Behaviour and Building Performance Group, Department of Architecture, University of Cambridge, Cambridge, United Kingdom, **2** Energy Policy Research Group, Judge Business School, University of Cambridge, Cambridge, United Kingdom

* rb867@cam.ac.uk

## Abstract

India locked down 1.3 billion people on March 25, 2020, in the wake of COVID-19 pandemic. The economic cost of it was estimated at USD 98 billion, while the social costs are still unknown. This study investigated how government formed reactive policies to fight coronavirus across its policy sectors. Primary data was collected from the Press Information Bureau (PIB) in the form press releases of government plans, policies, programme initiatives and achievements. A text corpus of 260,852 words was created from 396 documents from the PIB. An unsupervised machine-based topic modelling using Latent Dirichlet Allocation (LDA) algorithm was performed on the text corpus. It was done to extract high probability topics in the policy sectors. The interpretation of the extracted topics was made through a nudge theoretic lens to derive the critical policy heuristics of the government. Results showed that most interventions were targeted to generate endogenous nudge by using external triggers. Notably, the nudges from the Prime Minister of India was critical in creating herd effect on lockdown and social distancing norms across the nation. A similar effect was also observed around the public health (e.g., masks in public spaces; Yoga and Ayurveda for immunity), transport (e.g., old trains converted to isolation wards), micro, small and medium enterprises (e.g., rapid production of PPE and masks), science and technology sector (e.g., diagnostic kits, robots and nano-technology), home affairs (e.g., surveillance and lockdown), urban (e.g. drones, GIS-tools) and education (e.g., online learning). A conclusion was drawn on leveraging these heuristics are crucial for lockdown easement planning.

## Introduction

India locked down 1.3 billion people on March 25, 2020, in the wake of novel coronavirus COVID-19 pandemic. The Prime Minister of the country, Mr Narendra Modi, in his address to the nation on 24<sup>th</sup> March 2020, appealed to the nation that '. . . 21 days is critical to breaking the infection cycle. . . or else the country and your family could be set back 21 years. . .' [1]. In a sense, the government used the nudge of 'nationalism' as an effective measure to control the

**Data Availability Statement:** All media release files are available from the PIB database (https://pib. gov.in/AllRelease.aspx?MenuId=3)

**Funding:** RD received the Gates Cambridge Scholar by the Bill and Melinda Gates Foundation under the Grant Number OPP1144. The funders had no role in study design, data collection and analysis, decision to publish, or preparation of the manuscript.

**Competing interests:** The authors have declared that no competing interests exist.

disease spread. This nudge had critical public policy implications because it successfully convinced 1.3 billion people to abide by lockdown rules at high economic and social costs. The estimated economic cost of the Phase 1 lockdown of 21 days (March 25 to April 14, 2020) was estimated to be almost USD 98 billion [2]. Nudging is a design-based public policy approach which uses positive and negative reinforcements to modify the behaviour of the people. This approach has a high degree of subjectivity which makes it challenging to ascertain its reliability and replicability under public emergencies like pandemic, disaster, public unrest, etcetera [3]. Therefore, it is important to objectively untangle the nudges produced by government policies for efficiently handling national challenges like the COVID-19 pandemic.

Machine learning (ML) have proven to be a reliable technique in mining and distilling patterns in data and transform into predictive analytics for evidence-based policymaking. This technique is now widely used in deriving crucial information from big data into meaningful policy metrics. We have applied it to extract crucial nudges from official policy response and media releases of the GoI through its nodal agency—Press Information Bureau of India (PIB) [4]. This application of ML-based technique for nudge identification from government press releases defines the novelty of this study. The specific ML-technique employed in this study is called topic modelling (TM).

TM is a computational social science method that has its basis in text mining and natural language processing. It automatically analyses text data to determine cluster words for a set of documents [5]. TM has garnered significant importance in political science and rhetoric analysis [6]. Researchers have used TM to investigate reactions of different political communities on the same news for understanding political polarisation in the United States [7]. Similarly, in Korea, Kim & Jeong [8] have used TM on twitter dataset to analyse the temporal variation of the socio-political landscape of the 2012 Korean Presidential Election. In Germany, researchers have used a TM-approach to explore the multi-dimensionality of political texts and the discourses of public policies since National Elections of 1990 [9]. This study aided in understanding the polarising shifts in policy interventions that modulated the political narratives in Germany.

More recent applications of TM includes crisis identification in urban areas for evidence-based policymaking [10], deep narrative analysis for deriving intervention points for distributive energy justice in poverty [11] and informed public policy design in public administration [12]. However, none of the above applications of TM had explored the policy reactions of a government towards handling a national emergency using publicly available dataset. This study fills this gap while at the same time expands the application of data-driven textual analysis for analysis the reactiveness of public policies.

## Materials and methods

### Data collection and pre-processing

Data for this study were collected from the media releases of policies and plans of different ministries in the Press Information Bureau (PIB) platform [4]. English news and information with the keyword 'coronavirus', 'COVID', 'COVID-19' and 'nCoV' was collected and aggregated in a text format from January 15, 2020, and April 14, 2020. Manual filtering of the press and media releases based on the above keywords resulted in 396 documents from around 42 ministries of the Government of India. The entire text corpus from these documents consisted of 260,852 words. We classified these documents into 14 public policy categories, as illustrated in Table 1. Besides, we have also included COVID-19 briefings from the Prime Minister's Office in the policy categories (see Table 1).

**Table 1. Policy categories extracted from the ministries of the Government of India.**

| Sl. No. | Policy sectors | News and information from ministries |
|---|---|---|
| 1 | Agriculture and Food | Agriculture and Farmers Welfare; Fisheries, Animal Husbandry & Dairying; Food Processing Industries |
| 2 | AYUSH | Ayurvedic, Yoga and Naturopathy, Unani, Siddha and Homeopathy |
| 3 | Chemicals | Chemicals and Fertilizers; Commerce & Industry; Steel |
| 4 | Electronics & IT | Information & Broadcasting; Communications; Electronics & IT |
| 5 | Health | Health and Family Welfare |
| 6 | Home Affairs | Home Affairs; Defence; Finance |
| 7 | Labour & Commerce | Micro, Small & Medium Enterprises; Skill Development and Entrepreneurship; Textiles; Corporate Affairs; Personal, Public Grievances & Pensions |
| 8 | MHRD | Human Resource Development |
| 9 | PMO | Prime Minister's Office |
| 10 | Power | Power; Coal; Petroleum & Natural Gas; New & Renewable Energy |
| 11 | Science & Technology | Science and Technology; Statistics & Programme Implementation; |
| 12 | Social Justice | Rural Development; Social Justice & Empowerment; Tribal Affairs; Development of North-East Region; Minority Affairs; Panchayat Raj; Culture |
| 13 | Transport | Civil Aviation; Railways; Shipping; Tourism; Road, Transport and Highways |
| 14 | Urban | Housing and Urban Affairs; Environment, Forest and Climate Change Environment, Forest and Climate Change |

## Topic modelling using Latent Dirichlet Allocation (LDA)

Topic modelling (TM) refers to the task of identifying topics that best describes a set of documents. TM using Latent Dirichlet Allocation (LDA) algorithm is an unsupervised machine learning technique that automatically analyses text data to determine cluster words from a set of documents. It is based on the basic idea that each document can be expressed as a distribution of topics, and each topic can be described by a distribution of words [13]. The basic terminology used in LDA is based on the language of 'text collection', referring to entities such as "words", "documents" and "corpora". These terms are defined as (after [13]),

- A *word* is the basic unit of discrete data, defined to be an item from a vocabulary indexed by $\{1,\ldots,V\}$. We represent words using unit-basis vectors that have a single component equal to one and all other components equal to zero. Thus, using superscripts to denote components, the $v$th word in the vocabulary is represented by a $V$-vector $w$ such that $w^v = 1$ and $w^u = 0$ for $u \neq v$.

- A *document* is a sequence of $N$ words denoted by $\mathbf{w} = (w_1, w_2, \ldots, w_N)$, where $w_N$ is the $n$th word in the sequence.

- A *corpus* is a collection of M documents denoted by $D = \{\mathbf{w_1}, \mathbf{w_2}, \ldots, \mathbf{w_N}\}$.

The objective of TM is to extract latent semantic topics from large volumes of textual documents (i.e., corpora). LDA is a widely used unsupervised TM technique, with recent applications spanning across political science and rhetoric analysis [6–8, 14, 15], disaster management [10, 16, 17] and public policy [11, 12, 18]. It is a generative probabilistic method for modelling a corpus that assigns topics to documents and generates distributions over words given a collection of texts. Thus, providing a way of automatically discovering topics those documents contain. Fig 1 illustrates the probabilistic graphical model of LDA, and the

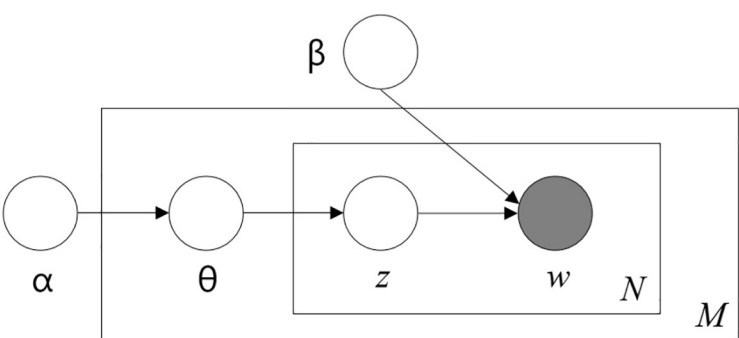

**Fig 1. Graphical model representation of LDA.** (Source: [13]).

probability calculation formula is illustrated in Eq 1:

$$p(D|\alpha, \beta) = \prod_{d=1}^{M} \int p(\theta_d|\alpha) \left( \prod_{n=1}^{N_d} \sum_{z_{dn}} p(z_{dn}|\theta_d)\, p(w_{dn}|z_{dn}, \beta) \right) d\theta_d \tag{1}$$

where, the boxes in Fig 1 are "plates" representing replicates. The outer plate represents documents (M), while the inner plate represents the repeated choice of topics (z) and words (w) within a document (N). 'Ө' is the topic distribution for document, i.e. 'α', 'β' are two hyperparameters of the Dirichlet distribution (see Eq 1). The third hyperparameter is the 'number of topics' that the algorithm will detect since LDA cannot decide on the number of topics by itself. We used our judgement to coarse estimate the total number of topics under each policy categories through a manually iterative process of reading the policy briefings. Following which the *ldatuning (v0.2.0)* package [19] in R (v3.5.3) was used to determine the number of topics in each of the topic models (discussed later in detail).

The analysis consisted of three main steps. The first step was the pre-processing of the documents by removing all the stop words (e.g., articles, such as "a," "an," and "the," and prepositions, such as "of," "by," and "from"), numbers, and punctuation characters and converted the text to lowercase in the corpora. And some general words appear in most of the government media releases like "name of ministers", "secretary", "union government" and courtesy words like "Shri", "honourable", "respected", "sir" and "thank you". We constructed a list of additional stop words that were colloquial terms in Indian-English and removed them from the text-corpus. This step is usually called lemmatisation [20]. Lemmatisation also involved removal of inflectional ending of words, and converting the grammatical form of a word into the base or dictionary form (known as Lemma) [20].

The second step was to fit the model using the lemmatised corpora. Using the *tidytext (v0.2.0)* package in the R programming language, we converted the article into a document-term-matrix (DTM) as per the specification of tidydata [21] rules. Each sentence was treated as a document in the DTM, that resulted in (*M*) unique documents that had *w* (words) and *z* (topics) as per LDA probability model specification (see Eq 1 and Fig 1). We adopted an iterative approach where we first specified the number of topics based on our judgement of the government's policy nudges and then tuned the appropriate number of topics as per the benchmarking metrics of Arun et al. [22], Cao et al. [23], Griffiths and Steyvers [24] and Deveaud et al., [25]. These metrics were part of the *ldatuning* package [19]; similar approach was also adopted by [12, 18]. We used the R package *topicmodels (v0.2–8)* to fit the LDA model [26].

The third step included visualisation and manual validation of the topics. For visualisation, we have used the *ggplot2 (v3.1.1)* package in R [27]. We have also estimated and visualised co-

occurrence of high-frequency keywords in the corpora using the methodology of Jan van Eck and Waltman [28]. The extracted topic was further analysed and interpreted concerning reactive policy steps using the epistemology of nudge theory in behavioural public policy [29].

## Evaluating topic models on nudge theory

Nudge theory is mainly concerned with the design of choices, which influences the decisions we make. It seeks to improve understanding and management of the 'heuristic' influences on human behaviour which is central to 'changing' people [30]. Epistemologically, Thaler and Sunstein [30] used nudge policies and interventions as an application of a conceptual framework called libertarian paternalism. The authors contend that retaining the freedom to choose is the best safeguard against a misguided policy intervention. The 'nudging' approach is paternalistic in the sense of motivating behaviour change that aligns with the target population's deliberative preferences [29]. Thus, libertarian paternalism relies on the assumption that each human being makes many decisions automatically and almost unthinkingly each day by following some innate rules of thumb [29]. It had been reported in literature that from a policy-instrumentation perspective, nudges constitute a less coercive form of government intervention compared to more traditional policy tools such as regulations and taxations [31]. While policy interventions can provide the right directions, it cannot suggest the promptness of the behaviour change. The behavioural nudge tactics, here, enable solving this last mile problem of policy intervention implementation success using "soft" techniques. Through this study, we wanted to understand how the Government of India used nudges as a public policy measure to fight the coronavirus outbreak.

# Results

## Topic co-occurrences

A keyword co-occurrence network was constructed with the 260,852 words dataset that shows a connected network of high-frequency words (see Fig 2). Words or terms that were mentioned at least 50 times in the text corpus was considered as high-frequency words. This threshold was decided based on the total number of unique words and the mode of its repetition in the text corpus. The co-occurrence representation has two components. Fig 2A illustrates a weighted network diagram of the high-frequency words. The weights were estimated based on the co-occurrences of a single word; the size of the bubble describes the relative weight associated with the words. Words like 'infection', 'virus', 'technology', 'testing', 'surveillance', 'passenger' and 'quarantine' had the highest weight and most interconnections, indicating the possible policy focus points during the early stage of the outbreak in India (between late-January to early-March). The general policy during this phase was on the containment of the cases. Extensive thermal screening of the passengers was conducted at the airports. During this stage, public policy was geared towards surveillance at the international borders. It remained a significant strategy until the national lockdown from March 24, 2020, until May 2020.

Similarly, Fig 2B illustrates the heat-map of the high-frequency words during the analysis period. The darker shades of grey in the heat-map indicate the policy points (or words) that had high frequency in the media briefs of the Government of India through the PIB. The darker shades of grey also illustrate higher co-occurrences of words in the text corpus. For example, 'coronavirus' → 'facility', 'effort': indicating policy efforts towards capacity building and healthcare facility management; 'coronavirus' → 'essential': extended focus on availing essential services during the lockdown period. 'mask'→ 'measure'; the use of masks had been extensively promoted as a COVID-19 control measure in India and currently made

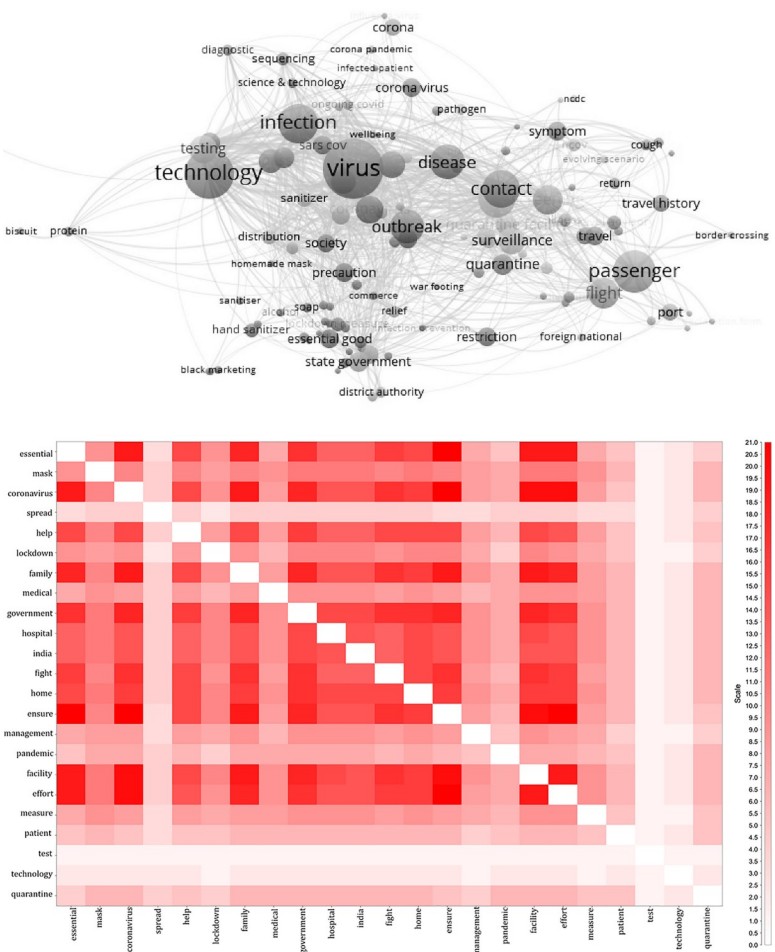

**Fig 2. High-frequency keyword co-occurrence representation on media briefings from Press Information Bureau (PBI) of the Government of India (GoI) in the wake of Covid-19 pandemic (mid-January 2020 to mid-April 2020).** Words that were repeated at least 50 times in the text corpus were considered in this analysis (n = 260,852): (a) Keyword co-occurrences map of high-frequency words. (b) Heat-map of high-frequency words.

compulsory by law. Similarly, 'lockdown' of 1.3 billion people of India has been the stringent public policy measure that has been enforced to curve the spread of coronavirus. The higher weighed words/policy measures with the lockdown can be seen in darker shades of grey in Fig 2B.

## Topic models

We have individually analysed the content of press releases from different ministries as per the policy classification presented in Table 1. In doing so, we estimated the approximate number of topic models for each of the policy categories using the benchmarking metrics of Arun2010 [22], CaoJuan2009 [23], Griffiths2004 [24] and Deveaud2014 [25], as illustrated in Table 2. The approximation of the number of topics was also made through judgement, where, we found that increasing the number of topics was affecting the interpretability of the topic models.

High-frequency words within the ministries are illustrated in Fig 3. The policies on agriculture and farmer's welfare focussed on ensuring food security and undisrupted supply chain

Table 2. Estimated topic models for each of the policy categories.

| Sl. No. | Policy sectors | Approximated number of topic models | Benchmarking criteria |
|---------|----------------|-------------------------------------|------------------------|
| 1 | Agriculture and Food | 2 | CaoJuan2009; Deveaud2014 |
| 2 | AYUSH | 2 | CaoJuan2009 |
| 3 | Chemicals | 2 | CaoJuan2009 |
| 4 | Electronics & IT | 4 | CaoJuan2009; Deveaud2014 |
| 5 | Health | 4 | CaoJuan2009; Deveaud2014 |
| 6 | Home Affairs | 10 | CaoJuan2009; Griffiths2004 |
| 7 | Labour & Commerce | 9 | CaoJuan2009 |
| 8 | MHRD | 4 | CaoJuan2009; Deveaud2014 |
| 9 | PMO | 8 | Deveaud2014 |
| 10 | Power | 3 | CaoJuan2009; Deveaud2014 |
| 11 | Science & Technology | 7 | CaoJuan2009; Griffiths2004 |
| 12 | Social Justice | 2 | CaoJuan2009; Deveaud2014 |
| 13 | Transport | 10 | CaoJuan2009 |
| 14 | Urban | 7 | CaoJuan2009 |

during the nationwide lockdown phase (see Fig 3). February to April is the harvesting time for winter crops in India that is crucial for food security in the country. In the wake of coronavirus and strict lockdown measures, the GoI allowed farmers to harvest. Besides, policy emphasis was laid on providing fiscal packages to the distressed farmers who were affected by national lockdown and supply chain disruption. Topic extraction through LDA (see Table 3) showed that the policy nudges were focussed on the continuity of harvest (topic 1, 'harvest', $\beta = 0.030$) and rerouting of the critical food supply chain (topic 2, 'lakh', $\beta = 0.100$) during the extended lockdown period for ensuring food security (topic 1, 'food security', $\beta = 0.150$).

AYUSH is an acronym for Ayurvedic, Yoga and Naturopathy, Unani, Siddha and Homeopathy. In the early stages of coronavirus pandemic in the country, this ministry released a series of press releases nudging people to follow the traditional medicinal practice of Ayurveda and maintaining good health and well-being through yoga (see Fig 3). The policy nudges, as revealed by the topic (see Table 3), showed a greater emphasis on increasing immunity through ayurvedic and herbal products. The topics also revealed higher stress on using Homeopathy ($\beta = 0.018$) and Ayurveda ($\beta = 0.032$) as preventive measure along with disciplined personal hygiene. It was observed that from the media releases that between January and the first week of March, AYUSH policies were aggressively nudging the use of traditional route to treat COVID-19. However, there was a shift in narrative during the mid-March as India experienced high infection rates. It focussed on promoting a healthy lifestyle through policy nudges using hashtags like #YOGAathome (see Fig 4).

The high-frequency word cloud for 'chemical' policy sector (see Table 1 and Fig 3) revealed higher policy stress on the availability of therapeutic drug and medical devices like ventilator and lifesaving equipment. Greater policy nudges were on empowering and motivating the manufacturing sector to contribute to medical device availability in the wake of coronavirus pandemic (see Fig 3). Three topic models were extracted that further expands on the policy nudges in this sector (see Table 3). Topic 1 indicates a greater emphasis on the bulk supply of medicine ($\beta = 0.065$) and contribution to the PM-CARES fund to ensure medicine availability in the country. Topic 2 further illustrates the aggressive nudging in manufacturing medical devices ($\beta = 0.048$). In addition, LDA extracts in Topic 3 revealed the higher impetus on supporting the frontline workers, see 'mask ($\beta = 0.048$)', 'PPE (personal protective equipment) ($\beta = 0.045$)', 'sanitiser ($\beta = 0.036$)' and 'drug surplus ($\beta = 0.030$) (see *Table 3*).

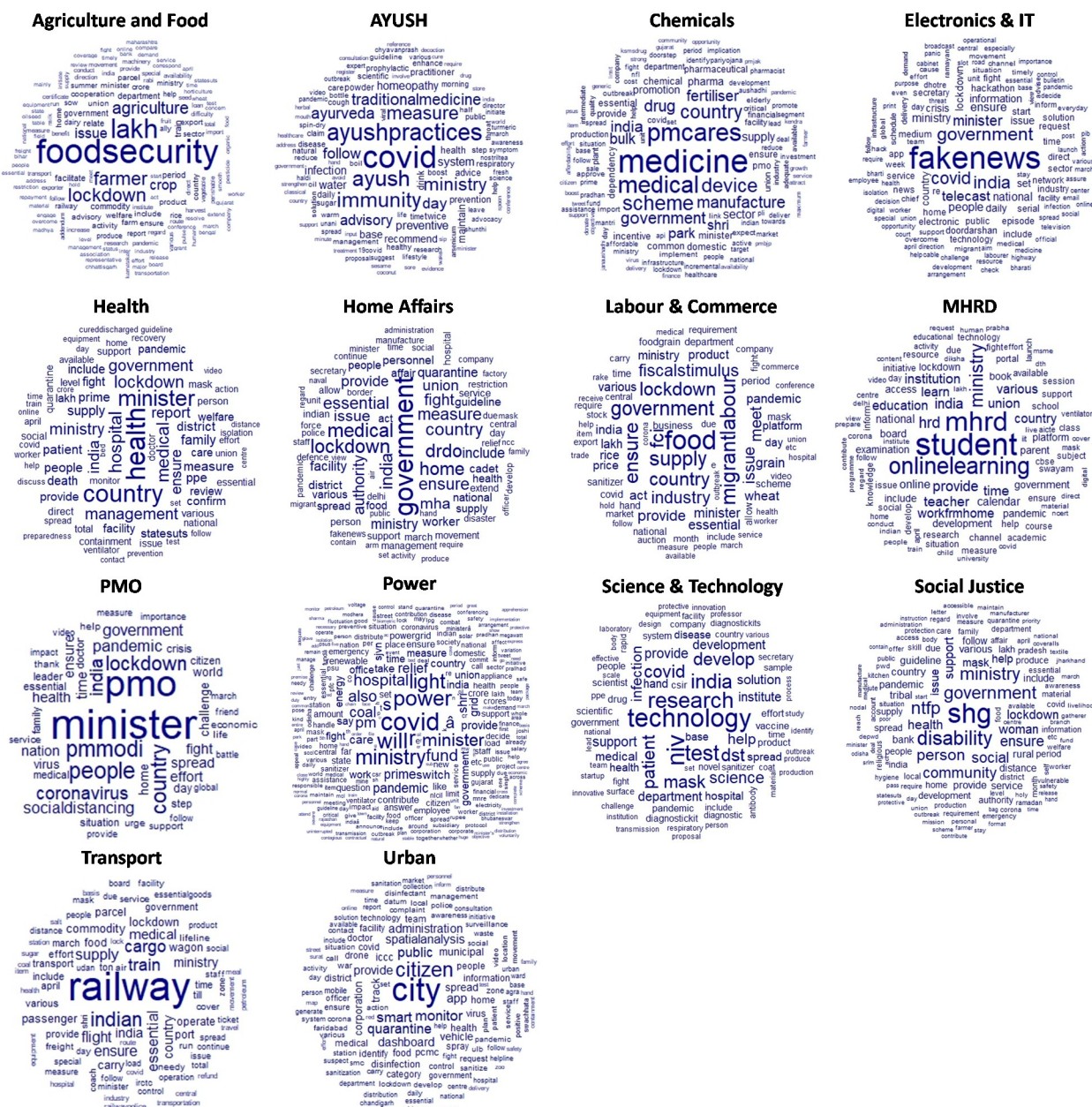

**Fig 3. High-frequency words in the official media releases of different ministries of the Government of India in the wake of COVID-19.**

The nudges from electronics and IT related policies were aggressive on tackling fake news in social media and keeping people indoors during the lockdown (see Fig 3). The repeated telecast of popular '80s and '90s TV shows were one of the distinct public policy nudges. It used nostalgia as a nudge to make the people conform to stay at home norm and practice social distancing measures [33]. These TV-shows ranged from family entertainer to religious and were broadcasted in the national channel called Doordarshan. Four topics were extracted (see Table 3), of which, topic 1 shows 'fake news' around COVID-19 as a high probability term (β = 0.070). It is being treated as a concern of national security. Topic 2 showed a similar discourse on guidelines concerning social media usage (β = 0.025) and fake news control (β =

**Table 3. Topic extracted by LDA as per the policy sectors.**

| Agriculture and Food | | | | | | |
|---|---|---|---|---|---|---|
| **Topic 1** | **Prob. (β)** | **Topic 2** | **Prob. (β)** | | | |
| Food security | 0.150 | Lakh | 0.100 | | | |
| Agriculture | 0.047 | Farmer | 0.059 | | | |
| Lockdown | 0.045 | Lockdown | 0.030 | | | |
| Crop | 0.040 | Issue | 0.025 | | | |
| Harvest | 0.030 | Agriculture | 0.018 | | | |
| **AYUSH** | | | | | | |
| **Topic 1** | | **Topic 2** | | | | |
| AYUSH practice | 0.070 | Traditional | 0.035 | | | |
| Covid-19 | 0.040 | Ayurveda | 0.032 | | | |
| Measure | 0.035 | Immunity | 0.030 | | | |
| Infection | 0.030 | Preventive | 0.029 | | | |
| Homeopathy | 0.018 | Hygiene | 0.032 | | | |
| **Chemicals** | | | | | | |
| **Topic 1** | | **Topic 2** | | **Topic 3** | | |
| Medicine | 0.065 | Device | 0.048 | Mask | 0.048 | |
| PM-CARES | 0.030 | Medicine | 0.045 | PPE | 0.045 | |
| Medical supply | 0.028 | Medical supply | 0.025 | Sanitiser | 0.036 | |
| Bulk supply | 0.020 | PM-CARES | 0.022 | Drug surplus | 0.030 | |
| Country | 0.018 | Government | 0.20 | India | 0.023 | |
| **Electronics & IT** | | | | | | |
| **Topic 1** | | **Topic 2** | | **Topic 3** | | **Topic 4** |
| Fake news | 0.070 | Ministry | 0.050 | Doordarshan | 0.025 | Fake news | 0.07 |
| Covid-19 | 0.05 | Fake news | 0.050 | Episodes | 0.020 | Security | 0.055 |
| Government | 0.045 | India | 0.048 | Information | 0.018 | People | 0.048 |
| Ensure | 0.038 | Social media | 0.025 | Crisis | 0.019 | Quarantine | 0.040 |
| Telecast | 0.015 | Media | 0.020 | Country | 0.015 | FCU | 0.025 |
| **MHRD** | | | | | | |
| **Topic 1** | | **Topic 2** | | **Topic 3** | | **Topic 4** |
| Student | 0.08 | Student | 0.079 | MHRD | 0.08 | Work-home | 0.040 |
| Online learning | 0.057 | Parent | 0.026 | Online learning | 0.034 | Online | 0.038 |
| Education | 0.044 | Time | 0.018 | Institution | 0.030 | Examination | 0.035 |
| Provision | 0.041 | India | 0.015 | Development | 0.028 | Schools | 0.020 |
| HRD | 0.028 | Nationwide | 0.010 | NBT | 0.025 | IIT | 0.015 |
| **Power** | | | | | | |
| **Topic 1** | | **Topic 2** | | **Topic 3** | | |
| PM-CARES | 0.015 | Stability | 0.017 | REP | 0.022 | |
| PSU | 0.013 | Light off | 0.015 | PM-CARES | 0.013 | |
| Covid | 0.011 | Renewable | 0.010 | MNRE | 0.010 | |
| Coal | 0.010 | Power | 0.008 | Grid | 0.007 | |
| Supply | 0.008 | Adequacy | 0.005 | Stability | 0.005 | |
| **Social Justice** | | | | | | |
| **Topic 1** | | **Topic 2** | | | | |
| Disability | 0.032 | Self Help Group | 0.007 | | | |
| Migrant worker | 0.027 | Woman | 0.025 | | | |
| Ministry | 0.025 | Lockdown | 0.023 | | | |
| Social security | 0.022 | NTFP | 0.022 | | | |
| Pandemic | 0.020 | Tribal | 0.022 | | | |

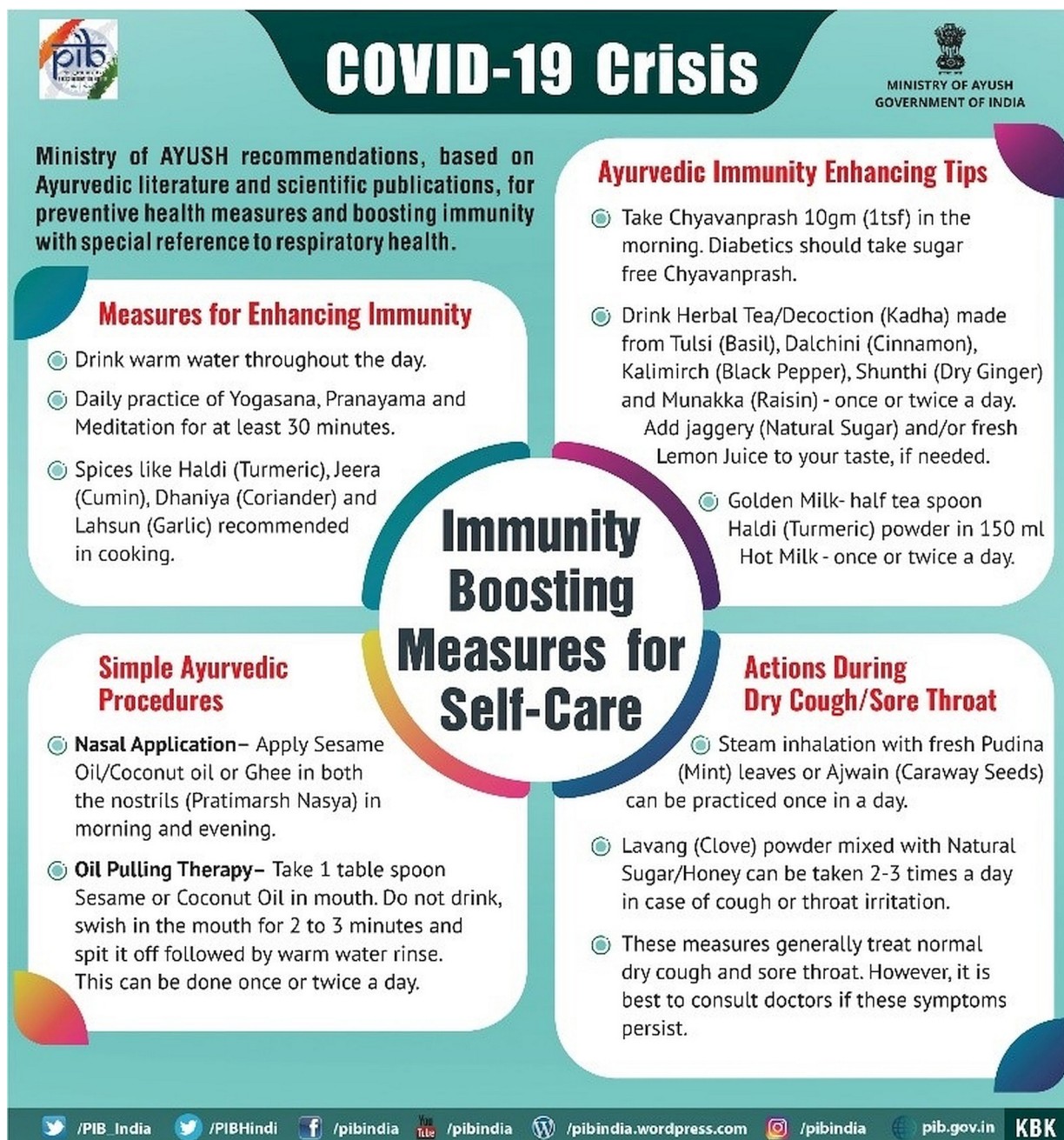

**Fig 4. AYUSH nudges on preventive health measures and boosting immunity.** (source: [32]).

0.050) through the Ministry of Electronics & IT. As aforementioned, India's public broadcaster DD aired '80s epic Hindu tale 'Ramayana' and 'Mahabharata' as a self-quarantine measure [33]. It is an application of nudging-based public policy measure referred to as the herd effect [29], illustrated in Topic 3 (see Table 3). Besides, various fiscal measures were taken to support the continuity of information flow through print and electronic media during the quarantine (β = 0.040) period (see Topic 4, Table 3). Fact-Checking Units (FCU) were set up to encourage the public to verify news for curbing fake news spread in social media (see Topic 4, Table 3).

In lines with the Electronics & IT, aggressive nudging on online learning was done by the Ministry of Human Resource Development (MHRD) (see Fig 3 and Table 3). Four topics were extracted, where online learning (Topic 1 and Topic 3) and work from home (Topic 4, β = 0.040) were the highest frequency words (see Table 3). The topic 1 illustrated the policy focus on infrastructure setup and provisioning of an online learning environment in the country. Subsequent nudging was done to the parents to encourage home-schooling by aggressively advertising the use of the National Digital Library of India, a GoI initiative under the National Mission on Education through Information and Communication Technology (NMEICT). This digital resource provides access to a multilingual virtual repository of learning resources across different levels of education with a single-window search facility (see Topic 2 and Topic 3). Policy impetus was on expanding the online educational resources by leveraging information and communication technologies (ICT). It was extracted on Topic 3, that shows 'institutional' (β = 0.030) and the National Book Trust (NBT) (β = 0.025). Hashtags like #StayHomeIndiaWithBooks were created as policy nudges by the NBT of MHRD, in its efforts to encourage people to read books while at home, is providing its select and best-selling titles for free download as part of its initiative. The NBT also launched a 'Corona Studies Series' to encourage readership of scientific books on COVID-19 to curb the spread of fake news. Similarly, policy nudges were made with #StayIN and #StayHome hashtags to encourage people to work from home (see Topic 4, β = 0.040).

The Ministry of Human Resource Development expansively nudged the start-up and innovation community in India to participate in the fight for COVID-19 by launching programs like 'Fight Corona IDEAthon. Moreover, Topic 4 also revealed the policy support provided on rescheduling national-level engineering and medical entrance examinations (β = 0.035). Besides extending the school and higher education lockdown period, the MHRD also converted public-owned school and university buildings into makeshift hospitals for COVID-19 patients.

Policy nudges in the power and energy sector were mostly dedicated to collecting funds for PM-CARES (see Fig 3). The extracted topics are illustrated in Table 3. Topic 1 consists of 'coal' as a high probable word (β = 0.010) that shows efforts in ensuring supply chain stability to thermal power plants in the country. Topic 2 further illustrates the concerns associated with the lockdown in the country with the 'lights off' request by the Prime Minister (PM) of India. The PM had nudged to the people to voluntarily switch off their lights for 10 minutes on April 5, 2020, as solidarity to frontline workers. It raised concerns of grid stability (β = 0.017) and power adequacy (β = 0.005). Power adequacy was also discussed through policy releases on renewable energy projects continuity even during the national lockdown (see Topic 4, Table 3). It can have a nudging impact on the post-COVID energy policies on decarbonisation and climate change mitigation.

Social justice in the wake of coronavirus pandemic is a critical policy focus point. Nudges included social security of migrant workers, labourers and women-led self-help group (see Fig 3). Guidelines were released for the person with a disability (see Table 3, Topic 1, β = 0.032) and migrant workers stuck in cities amidst the nationwide lockdown (β = 0.027). Topic 2 further illustrates the social protection policies for the tribal communities. They were affected by the national lockdown and its impact on their livelihood-based on Non-Timber Forest Products (β = 0.022). Special fiscal packages were planned for the self-help group (SHG) run by rural women (see Topic 2, β = 0.022).

Ministry of Home Affairs and Ministry of Defence are the institutions that deal with national security and peacekeeping. In this study, we combined the press releases of both the ministries as 'Home Affairs' (see Table 1) as they have been working in tandem governing the national lockdown rules in the wake of coronavirus pandemic. Fig 3 shows the high-frequency

**Table 4. Topic extracted by LDA for Home Affairs.**

| | | | | | | | |
|---|---|---|---|---|---|---|---|
| **Home Affairs** | | | | | | | |
| **Topic 1** | **Prob. (β)** | **Topic 2** | **Prob. (β)** | **Topic 3** | **Prob.(β)** | **Topic 4** | **Prob.(β)** |
| India | 0.140 | SupplyChain | 0.100 | Government | 0.080 | Lockdown | 0.082 |
| Cadet | 0.110 | AirDrop | 0.059 | Lockdown | 0.070 | Surveillance | 0.065 |
| Support | 0.060 | Medical | 0.030 | National | 0.065 | Drones | 0.060 |
| Border | 0.055 | AirForce | 0.025 | Restrictions | 0.050 | DRDO | 0.040 |
| DRDO | 0.053 | Defence | 0.018 | Surveillance | 0.045 | Containment | 0.034 |
| **Topic 5** | **Prob. (β)** | **Topic 6** | **Prob. (β)** | **Topic 7** | **Prob.(β)** | **Topic 8** | **Prob.(β)** |
| Spread | 0.075 | Provide | 0.121 | Tablighi | 0.170 | Airport | 0.160 |
| Hand | 0.065 | SupplyChain | 0.110 | Delhi | 0.152 | Surveillance | 0.070 |
| Person | 0.061 | Governance | 0.093 | Spike | 0.100 | Borders | 0.055 |
| Devices | 0.057 | Essentialitems | 0.073 | Lockdown | 0.090 | Passenger | 0.053 |
| PPE | 0.050 | Railways | 0.050 | Religious | 0.051 | Checkpoint | 0.050 |
| **Topic 9** | **Prob. (β)** | **Topic 10** | **Prob. (β)** | | | | |
| DRDO | 0.100 | Defence | 0.098 | | | | |
| R&D | 0.066 | Hospital | 0.064 | | | | |
| Masks | 0.060 | Ventilator | 0.061 | | | | |
| PPE | 0.056 | Navy | 0.060 | | | | |
| CriticalCare | 0.053 | Airforce | 0.056 | | | | |

words from the home affairs. It exhibited ensuring the supply of essential commodities, ensuring lockdown governance, surveillance measures and quarantine facilities as highlighted words. Ten topic models were extracted using LDA, as illustrated in Table 4.

Topic 1 (see Table 4) illustrates the actions taken by the Indian defence in increasing surveillance of the borders (β = 0.055) and the involvement of Defence Research and Development Organisation (DRDO) (β = 0.053). Similarly, topic 2 shows the involvement of the Indian Air Force (IAF) (β = 0.025) in ensuring the supply chain (β = 0.100) of essential items amidst national lockdown. IAF planes were used to transport medicines, PPE, masks and life-saving devices across the nations (see Table 4). The Ministry of Home Affairs (MHA) is the decision-making body on ensuring lockdown and national security are maintained in the wake of the pandemic. Nudging was around extensive surveillance and ensuring public follow the restrictions (see Topic 3, Table 4). The DRDO was also involved in extensive research and development of containment equipment (β = 0.034). It included scaling up the technology for the use of aerial drones for surveillance (β = 0.065). There was also extensive use of spatial mapping technologies for contact tracing amidst national lockdown.

The MHA was also extensively involved with the manufacturing sector to design and develop low-cost ventilators, PPE, sanitisers and masks (see Topic 5 and Topic 6, Table 4). Extensive nudging was done to ensure that the government was actively involved in delivering essential items by engaging with the supply chain of Indian Railways (see Topic 6). Moreover, amidst the national lockdown, spikes in coronavirus cases were observed in New Delhi due to religious gathering (Tablighi Jamaat congregation), the MHA had to tighten up surveillance and increase the nationwide contact tracing (see Topic 7). This event was speculated as to India's worst coronavirus vector [34].

Besides, MHA ensured surveillance at the airports and international borders and were the first responders during the early stage of the pandemic in the country (see Topic 8). It used nudging at the airport to ensure travellers maintain a 14-day home quarantine by stamping people with 'Home Quarantine 'on forearms (see Fig 5).

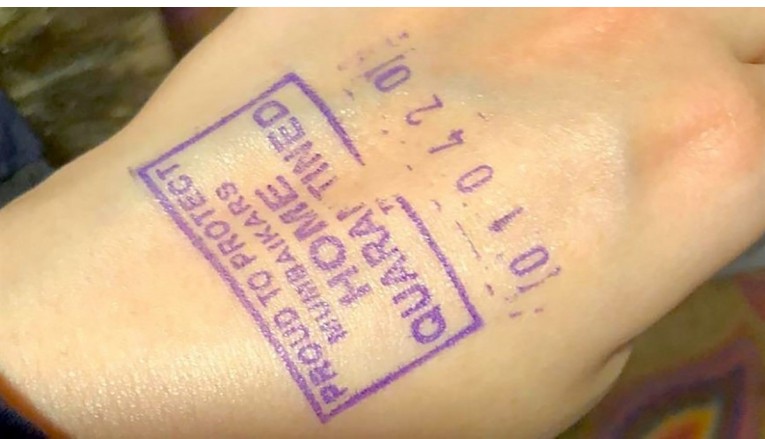

**Fig 5. 'Home quarantined' stamp for travellers as nudging for self-isolation.** (source: [35]).

Furthermore, Topic 9 and Topic 10 (see Table 4) indicated the efforts made in ensuring the availability of critical care infrastructure and PPE in remote parts of the country using National Cadet Corps (NCC). The National Cadet Corps, a Tri-Services Organisation, comprising the Army, Navy and Air Wing, engaged in grooming the youth of the country into disciplined and patriotic citizens. The cadets were deployed for various duties like traffic management, supply chain management, preparation and packaging of food items, distribution of food and essential items, queue management, social distancing, operating control centres and CCTV control rooms. Besides, NCC cadets were sensitising the public against COVID-19 by sending messages (as nudges) on social media platforms like Twitter, Instagram and WhatsApp, etcetera. They further enhanced the mental and social protection of migrant workers and people living in hyperdense settlements like slums by leveraging ICTs [36]. Besides, the MHA worked closely with the Ministry of Finance to plan 'Economic Distress Relief Package' that involves instant relief in the form of providing a slew of measures that will ensure food grain and other essential as well as financial assistance to disadvantaged sections of the society.

The surveillance in urban areas was done using smart technologies (see Fig 3) that included drones, spatial analysis, low-power Bluetooth mobile phone applications and humanoid robots [37]. The Smart City program of India [38] has been leveraged as critical vantage points in the COVID-19 fight by the Ministry of Housing and Urban Affairs (MoHUA) [39]. For example, helium balloon attached with cameras for surveillance on lockdown violators were used in the Vadodara Smart City, Gujarat. A COVID-19 War Room at Bengaluru was established to enable real-time data-driven decision-making using a single dashboard. Similarly, tele-video consultation facilities were launched in Agra to enable E-Doctor Service for the local population [39]. See Table 5 for the topics extracted by LDA concerning urban policies.

The significant policy nudges were on requesting the public to comply with the strict quarantine rules using drones and smart surveillance technologies (see Table 5 and Fig 6A). Nudging was also on the use of COVID-19 contact tracing apps, and GIS-based methods for monitoring quarantined public at a municipality level. Special attention was given to the routine solid waste collection, transportation and disposal activities along with cleaning and scrapping were carried out efficiently to keep the cities clean. In few highly dense urban centres, disinfection tunnels were installed (see Fig 6B) with facilities of thermal screening by taking temperature. Pedestal operated hand-wash and soap dispenser, mist spray of sodium hypochlorite solution and hand dryer facility. The topic extracted in Table 6 compiles all these measures to control the spread of COVID-19.

**Table 5. Topic extracted by LDA concerning urban sector.**

| Urban | | | | | | | |
|---|---|---|---|---|---|---|---|
| Topic 1 | Prob. (β) | Topic 2 | Prob. (β) | Topic 3 | Prob.(β) | Topic 4 | Prob.(β) |
| Virus | 0.050 | Smart | 0.021 | Quarantine | 0.028 | Various | 0.030 |
| Track | 0.030 | Technology | 0.015 | Monitor | 0.035 | Technologies | 0.040 |
| Spread | 0.070 | Surveillance | 0.030 | Dashboard | 0.030 | Public | 0.030 |
| Public | 0.040 | Control | 0.048 | Citizen | 0.040 | Identify | 0.045 |
| Monitor | 0.050 | City | 0.180 | App | 0.100 | City | 0.040 |
| Topic 5 | Prob. (β) | Topic 6 | Prob. (β) | Topic 7 | Prob.(β) | | |
| Vehicles | 0.075 | Sanitise | 0.030 | Track | 0.004 | | |
| Spatial analysis | 0.065 | Monitor | 0.036 | Smart | 0.003 | | |
| Municipal | 0.061 | Mobile | 0.040 | GIS | 0.003 | | |
| Government | 0.057 | Essential items | 0.050 | Near-me | 0.005 | | |
| City | 0.050 | Citizens | 0.110 | Urban | 0.003 | | |

The transportation sector played a critical role in maintaining the supply chain of essential items. Fig 3 shows the high-frequency words in the transportation sector that includes freight transport, railways, shipping and road and highways. The topics extracted by LDA is illustrated in Table 6 with the policy nudges in the transportation sector in the wake of coronavirus pandemic in India.

In the wake of coronavirus, the Government of India consistently nudged the scientific community of India to fight the pandemic by launching a series of funding through the Department of Science and Technology (DST). Policy design relied on evidence-based decision-making. High-frequency keywords concerning Science and Technology (S&T) sector is illustrated in Fig 3. The topics extracted by LDA on S&T is illustrated in Table 7.

The National Institute of Virology (NIV) was at the forefront of testing, which provided the technical guidance for testing labs across the country (see Table 7). Academic and research institutions were encouraged to submit competitive interdisciplinary research proposals to focus on the development of affordable diagnostics, vaccines, antivirals, disease models, and other R&D to study COVID-19 (see Table 7).

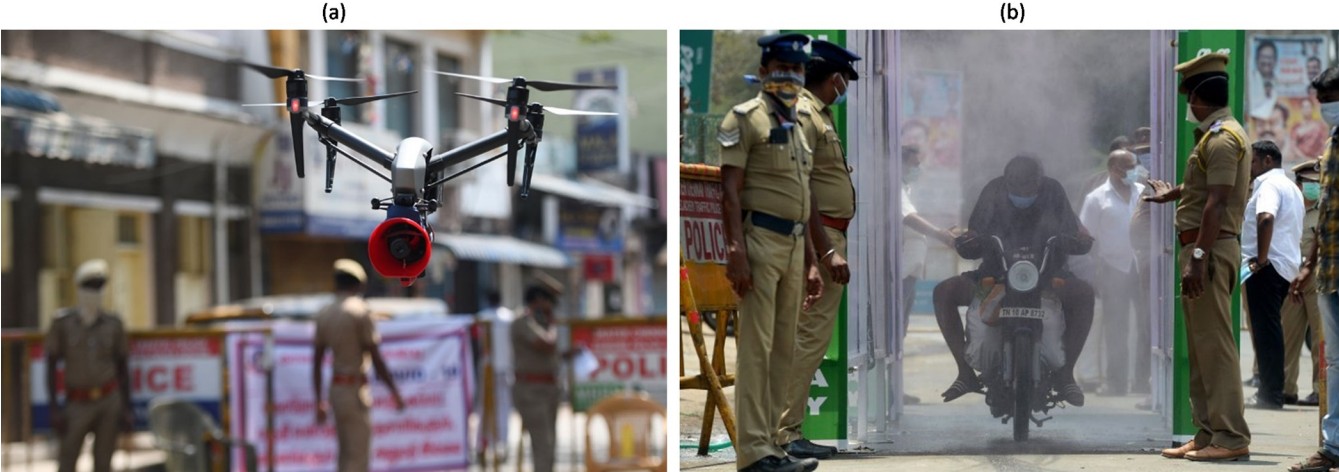

**Fig 6.** (a) A drone used by police to monitor activities of people and spread awareness announcements; (b) A motorist rides through a disinfection tunnel. (source: [39, 40]).

**Table 6. Topic extracted by LDA in the transport sector.**

| Transport | | | | |
|---|---|---|---|---|
| Topic 1 | Prob. (β) | Topic 2 | Prob. (β) | Policy nudges |
| India | 0.160 | Train | 0.100 | Indian railways played a major role in ensuring supply chain of essential items are operated in business-as-usual conditions amidst the national lockdown. Extensive nudging was done by the Ministry of Railways ensuring public that all there won't be any food or medicine shortage. It was repeatedly done to reduce public hysteria and mass panic. |
| Lockdown | 0.150 | People | 0.059 | |
| Supply | 0.100 | Railway | 0.030 | |
| Medical | 0.050 | SupplyChain | 0.025 | |
| Railway | 0.080 | Lockdown | 0.018 | |
| Topic 3 | | Topic 4 | | |
| Issue | 0.070 | Railway | 0.160 | Nudges were by the railway factories where they started to produce PPE and masks to curb national shortage for frontline workers. Railways rerouted several long-distance trains to support remote hospitals with lifesaving equipment and PPE. |
| Needy | 0.065 | Commodity | 0.060 | |
| Zone | 0.055 | Masks | 0.050 | |
| Health | 0.050 | PPE | 0.050 | |
| Hospital | 0.048 | Factory | 0.040 | |
| Topic 5 | | Topic 6 | | |
| Supply | 0.095 | Passenger | 0.100 | During lockdown, supply chain of essential commodities was maintained through freight carriers by road and railways. Old trains were converted to isolation wards using frugal innovation in the wake of exponentially rising coronavirus cases (see Fig 10). |
| Freight | 0.090 | Effort | 0.100 | |
| Lockdown | 0.075 | Wagon | 0.080 | |
| Load | 0.060 | Makeshift | 0.076 | |
| Carry | 0.050 | Cabin | 0.060 | |
| Topic 7 | | Topic 8 | | |
| UDAAN | 0.200 | Flights | 0.250 | The Ministry of Civil Aviation operating cargo planes with passenger airlines, Indian Navy and Indian Airforce to deliver medicine and testing kits to remotest part of the country. Public nudging was done through the term 'Lifeline UDAAN' indicating the aforementioned flights supplying essential items. |
| Flights | 0.100 | Provide | 0.050 | |
| Emergency | 0.055 | Navy | 0.030 | |
| AirForce | 0.040 | Emergency | 0.020 | |
| Medical | 0.030 | Shelter | 0.011 | |
| Topic 9 | | | | |
| Port | 0.110 | | | The Ministry of Shipping started 100% surveillance by installing thermal screening, detection and quarantine systems immediately for disembarking Seafarers or Cruise Passengers. Safety procedures were made compulsory while handling cargo at ports. |
| Shipping | 0.100 | | | |
| Cargo | 0.075 | | | |
| Covid | 0.070 | | | |
| Screening | 0.056 | | | |
| Topic 10 | | | | |
| Railway | 0.300 | | | Railways nudged public by issuing 100% refund on cancellation of tickets to discourage travel in the wake of covid-19 pandemic. In addition, they removed all senior citizen benefits and concessions to discourage ticket sales. |
| Refund | 0.080 | | | |
| Tickets | 0.050 | | | |
| Senior | 0.045 | | | |
| Citizen | 0.030 | | | |

Scientific innovation during this period includes robots for encouraging social distancing in public spaces and healthcare centres (see Fig 7). A contact tracing app (AarogyaSetu) using GPS and Bluetooth to inform people when they are at risk of exposure to COVID-19. Low-cost, easy-to-use, and portable ventilators that can be deployed even in rural areas of India. To nudge people into using the application was provided by frequent reminders through SMS. Innovations were also done in ensuring public-space hygiene through the development of water-based sanitiser disinfectant and technology to dispenses ionised water droplets to oxidise the viral protein [42]. The DST set up a task force to map technologies developed by start-ups related to COVID-19. It is funding start-ups to develop relevant innovations such as rapid

**Table 7. Topic extracted by LDA for the Science and Technology (S&T) sector.**

| Science and Technology | | | | |
|---|---|---|---|---|
| **Topic 1** | **Prob. (β)** | **Topic 2** | **Prob. (β)** | **Policy nudges** |
| Develop | 0.110 | Test | 0.150 | Policy nudges were on the development of affordable rapid testing kits, PPE, medical devices and infection preventive technologies. |
| Health | 0.070 | Mask | 0.120 | Office of the Principal Scientific Advisor to the Government of India extensively nudged public to adopt homemade masks and adhere to frequent hand washing to curb the spread of coronavirus [41]. |
| Facility | 0.050 | DST | 0.040 | |
| Time | 0.030 | Patients | 0.050 | |
| Rapid | 0.028 | Hand wash | 0.060 | |
| **Topic 3** | | **Topic 4** | | |
| India | 0.160 | Technology | 0.160 | Private sector R&D institutions and industry were nudged to join the fight against covid-19. Government urged them to develop highly scalable technologies and testing kits to ramp up national testing. Use of technology to ensure strict lockdown was also nudged by DST. |
| Covid | 0.080 | NIV | 0.050 | |
| Technology | 0.040 | Hospital | 0.050 | |
| Effort | 0.040 | Help | 0.030 | |
| Industry | 0.040 | Proposal | 0.030 | |
| **Topic 5** | | **Topic 6** | | |
| Research | 0.120 | Science | 0.120 | R&D of low-cost rapid testing kits were nudged from the Office of the Principal Scientific Advisor to the Government of India. Extensive nudging was also done to the public to install a government approved contact tracing app called 'AarogyaSetu'. Call for research proposals and innovation challenges were launched to fight coronavirus. |
| Solution | 0.090 | AarogyaSetu | 0.090 | |
| Diagnostickit | 0.050 | Innovation | 0.070 | |
| Patient | 0.050 | Funding | 0.050 | |
| Covid | 0.030 | Vaccine | 0.030 | |
| **Topic 7** | | | | |
| Infection | 0.110 | | | The DST also nudged micro, small and medium scale industries (MSMEs) and rural enterprises to produce large-scale PPE and masks. It was done to ramp up the PPE, masks and sanitiser production in rural areas that could keep the economy running. |
| Research | 0.060 | | | |
| Development | 0.060 | | | |
| PPE | 0.040 | | | |
| Masks | 0.030 | | | |

testing for the virus (see Table 7). The national government launched the COVID-19 solution challenge on March 16 that invited innovators to offer ideas and solutions for tackling the pandemic. It was a policy nudge on crowdsourcing ideas that encouraged public and the start-up ecosystem to contribute to this fight. BreakCorona is one such crowdsourced initiative that received 1,300 ideas and 180 product solutions within two days of launch [42]. An online

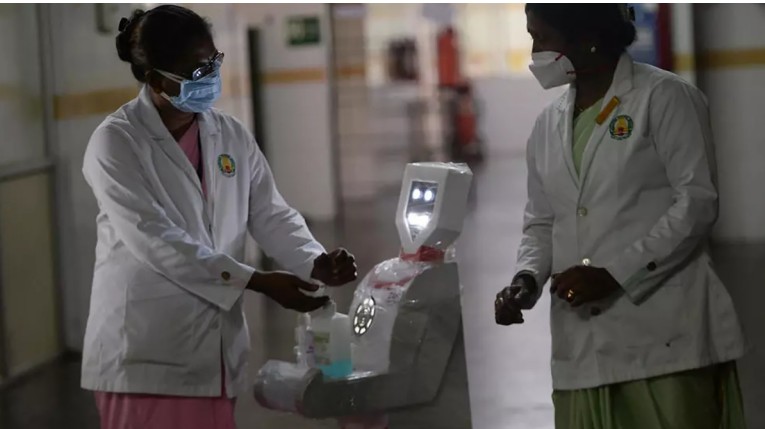

**Fig 7. Robot dispensing sanitiser in isolation wards in a hospital in Chennai, India.** (source: [43]).

**Table 8. Topic extracted by LDA for health sector.**

| Health | | | | | | | |
|---|---|---|---|---|---|---|---|
| **January** | | | | | | | |
| Topic 1 | Prob. (β) | Topic 2 | Prob. (β) | Topic 3 | Prob.(β) | Topic 4 | Prob.(β) |
| Report | 0.035 | Preparedness | 0.032 | Health | 0.058 | Health | 0.040 |
| China | 0.022 | Report | 0.023 | Ministry | 0.058 | Screening | 0.020 |
| nCoV | 0.021 | Travel | 0.020 | Airport | 0.040 | Restriction | 0.020 |
| Passenger | 0.020 | Review | 0.020 | Review | 0.030 | Airport | 0.018 |
| Airport | 0.019 | Ministry | 0.020 | Screening | 0.020 | China | 0.015 |
| **February** | | | | | | | |
| Topic 1 | | Topic 2 | | Topic 3 | | Topic 4 | |
| Health | 0.070 | Passenger | 0.045 | Advisory | 0.043 | Travel | 0.045 |
| Travel | 0.050 | Health | 0.042 | Government | 0.040 | Restrictions | 0.040 |
| China | 0.050 | State/UT | 0.038 | Screen | 0.035 | Passenger | 0.032 |
| Welfare | 0.040 | Airport | 0.032 | Flights | 0.030 | Surveillance | 0.031 |
| Screening | 0.030 | nCoV | 0.030 | nCoV | 0.028 | International | 0.030 |
| **March** | | | | | | | |
| Topic 1 | | Topic 2 | | Topic 3 | | Topic 4 | |
| India | 0.052 | Passenger | 0.065 | Country | 0.070 | Test | 0.054 |
| Health | 0.051 | Screening | 0.050 | Ministry | 0.055 | PPE | 0.050 |
| Safety | 0.040 | Travel | 0.050 | Health | 0.042 | Healthcare | 0.045 |
| Hospital | 0.038 | Restriction | 0.040 | ICMR | 0.040 | Management | 0.040 |
| Preparedness | 0.020 | Health | 0.038 | Research | 0.038 | Lockdown | 0.035 |
| **April** | | | | | | | |
| Topic 1 | | Topic 2 | | Topic 3 | | Topic 4 | |
| Ministry | 0.080 | Management | 0.075 | Health | 0.075 | Country | 0.080 |
| Lockdown | 0.048 | PM | 0.038 | Briefing | 0.063 | Masks | 0.030 |
| Social distancing | 0.045 | Essential | 0.035 | PPE | 0.030 | Pandemic | 0.025 |
| Hospital | 0.030 | Medicine | 0.030 | Testing | 0.025 | Testing | 0.022 |
| Testing | 0.028 | Testing | 0.025 | Quarantine | 0.020 | Hygiene | 0.020 |

crowdsourced portal called Coronasafe-Network, was also set-up by volunteers to provide real-time open-source, public platform containing details on COVID-19 precautions, tools and responses which serves as a useful starter-kit for innovators [42].

Table 8 shows the topic extracted by LDA in the health sector between January to April. The results show that in January, the policy nudges were in evaluating the risk of incoming travellers coming from China and extending surveillance at international airports. High-frequency words associated with such nudges can be seen in Fig 8. The change in policy narratives of the health ministry can be seen with the spread of infection in the country (see February, Table 8). The nudges were on enhancing thermal screening at airports of international arrivals and imposing travel restriction (see Fig 8).

Furthermore, topics extracted for February also indicates the beginning phase off restrictions such as advisory on social distancing and frequent hand washing as a possible preventive measure of towards COVID-19 infection (see Table 8). In additions, the Ministry of Health & Family Welfare (MoHFW) began extensive nudging states and union territories of India to follow norms on social distancing and thermal screening of international travellers. More travel restrictions were imposed for China, Iran, Spain and Italy.

By March, the policy narratives shifted to imposing hard restrictions on travel, and people were discouraged from visiting crowded and public spaces. Strict social distancing nudges

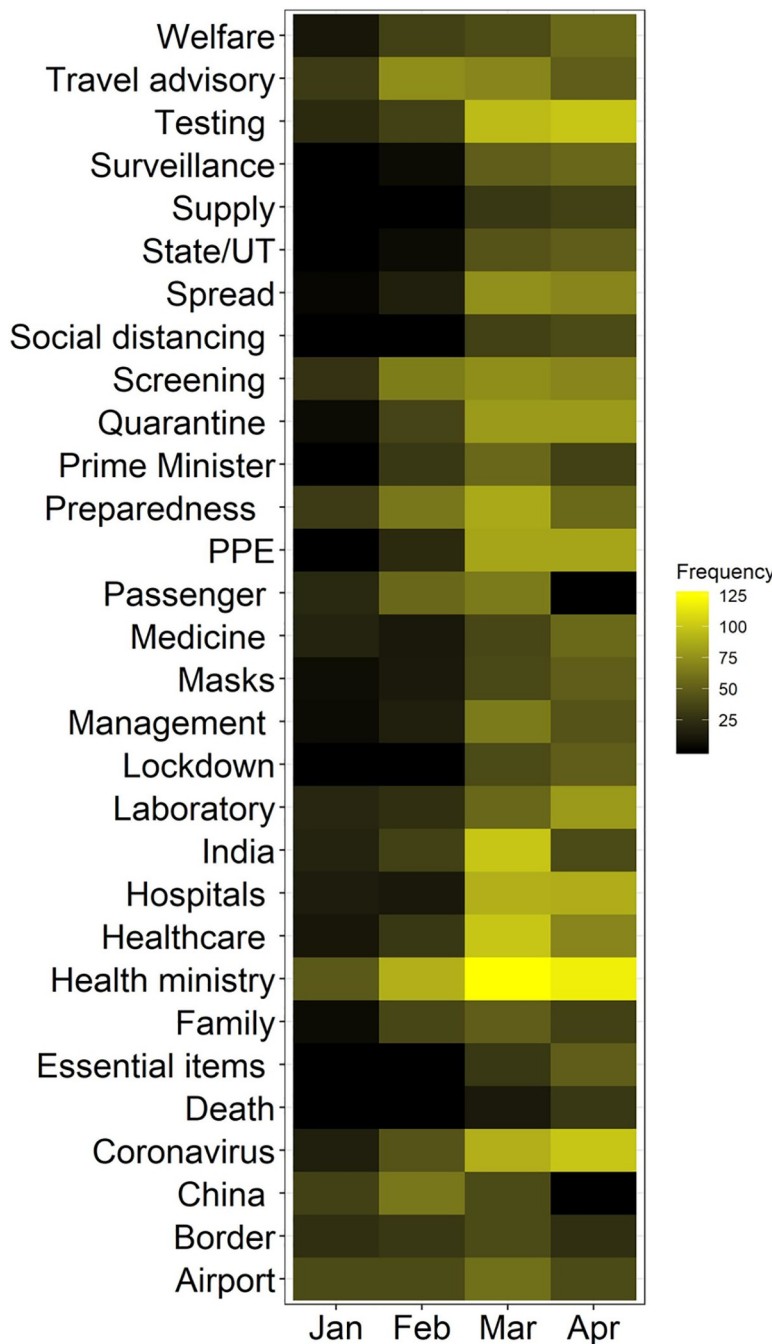

**Fig 8. Temporal high-frequency word dynamics in the health policy in the wake of COVID-19 in India.**

were being imposed as reactive policy. At the same time, MoHFW began to increase testing capacity across the country and on March 25, 2020, Phase 1 of lockdown began. People were nudged constantly during this phase to strictly adhere to the lockdown rules, use masks and wash hand frequently. Manufacturing units were asked to produce PPE, hand sanitiser and masks to meet the national demand (see Table 8, March). The Indian Council of Medical Research (ICMR) was the nodal agency for coordinating with press and MoHFW concerning

the development regarding COVID-19 pandemic. It started daily briefing on government policies and preparedness on fighting coronavirus (see March, Table 8 and Fig 8).

The policy nudges for April was centred towards strengthening the COVID-19 specific healthcare requirements. Increasing the number of testing done per 1000 people was one of the significant agenda along with the social distancing measures. This phase was also marked by innovation in indigenous science and technology for empowering frontline working to fight COVID-19 (see Tables 7 and 8). During this period, policy nudges were also towards ensuring food security and availability of essential items and medicines across the nation (see Fig 8). Masks were made compulsory at public spaces across the nation (see Table 8, April and Fig 8).

Prime Minister's Office (PMO) was at the forefront of the fight against coronavirus, the high-frequency words are illustrated in Fig 9. Prime Minister Narendra Mod's nudges were driving the COVID preparedness, action and mitigation strategies in the country. His frequent public appearance was the most significant factor that created nudges in keeping a country of 1.3 billion people under strict lockdown and social distancing measures (see Table 9). In this process, the PMO spearheaded the creation of 'Prime Minister's Citizen Assistance and Relief in Emergency Situations Fund' (PM CARES Fund) for dealing with emergency or distress situation like posed by Covid-19 pandemic. PM-CARES was created to nudge the public into micro-donations and show the strength of public participation to mitigate any issue. Most of the nudges were in the form of social media advertisements, SMS forwards and repeated reminders through broadcasting media.

The PMO was created 'Covid-19 Economic Response Task Force' to deal with the economic challenges caused by the pandemic. Prime Minister (PM) also nudged the business community and higher-income groups to look after the economic needs of those from lower-

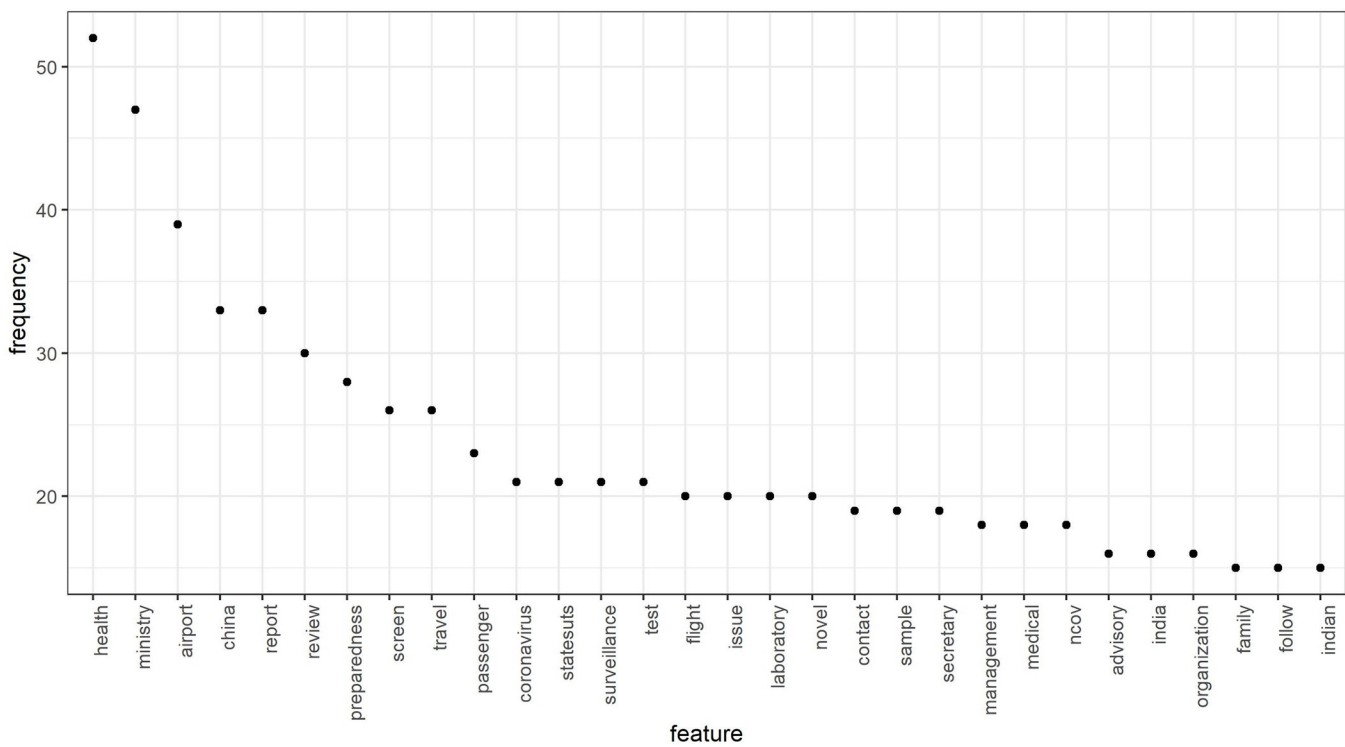

**Fig 9. Keyword distribution for the Prime Minister's Office (PMO), Government of India.**

**Table 9. Topic extracted by LDA for Prime Minister's Office.**

| | | | | Prime Minister's Office |
|---|---|---|---|---|
| **Topic 1** | **Prob. (β)** | **Topic 2** | **Prob. (β)** | **Policy nudges** |
| Country | 0.250 | Step | 0.100 | PM nudged the citizens of India to join the fight against coronavirus and stand up to the challenge in solidarity with the frontline workers. The nudges were on building mental strength as an Indian citizen and set examples of humanity. |
| PM Modi | 0.160 | Citizen | 0.080 | |
| India | 0.150 | Family | 0.079 | PM urged the people of India to be proud of the frontline workers and support the cause by donating it to PM-CARES. |
| Coronavirus | 0.080 | Challenge | 0.075 | |
| Ministry | 0.050 | Medical | 0.060 | |
| **Topic 3** | | **Topic 4** | | |
| PMO | 0.160 | People | 0.160 | PM assured the nation on economic policy in challenging times. He announced a USD 23 billion support measure for the welfare of the people of India. His nudges were also to higher income group and the business community to the economic needs of those from lower-income groups. |
| Economic | 0.080 | World | 0.050 | |
| Crisis | 0.040 | Meet | 0.050 | In this effort, PM also nudged the leaders from various countries, especially SAARC, G-20, and BRICS nations to contribute to joint funding for fighting covid019 in South-East Asia. |
| Impact | 0.040 | Fund | 0.030 | |
| PM-CARES | 0.040 | International | 0.030 | |
| **Topic 5** | | **Topic 6** | | |
| Pharma | 0.120 | AYUSH | 0.120 | PM nudged the pharma sector to maintain a regular supply of medicines and medical equipment. He urged the pharma industry and the broader scientific community of India to work on diagnostic kits and possible vaccine agent for coronavirus. |
| Solution | 0.090 | AarogyaSetu | 0.090 | |
| Science | 0.050 | Handwash | 0.070 | He nudged them that it is essential to maintain the supply of essential medicines and prevent black marketing and hoarding. |
| Pandemic | 0.050 | Mask | 0.050 | Besides, PM also nudged the general public to install the AarogyaSetu contact tracking app and improve wellbeing and general immunity through Ayurveda. He promoted #YogaAtHome to de-stress the mind and strengthen the body during this challenging phase. |
| Innovation | 0.030 | Wellbeing | 0.030 | |
| **Topic 7** | | **Topic 8** | | |
| Social distancing | 0.080 | Frontline | 0.200 | PM thanked the medical fraternity for its selfless service to the nation as the frontline defence against coronavirus. He assured that the security of frontline workers is of utmost importance, and the government will take every step to protect them. |
| Lockdown | 0.060 | Medical | 0.160 | |
| Fight | 0.150 | Workers | 0.110 | He nudged the state and union territories to step up their governance in the fight towards COVID-19 by assuring strict lockdown and social distancing measures. |
| State/UT | 0.090 | Security | 0.050 | |
| Governance | 0.070 | Law | 0.050 | |

income groups, from whom they take various services, urging them not to cut their salary on the days they are unable to render the services due to inability to come to the workplace. PM stressed on the importance of humanity during such times [44]. The topics extracted by LDA on PMO is illustrated in Table 9.

## Discussion

We studied the reactive public policies in India in the wake of coronavirus pandemic through topic modelling using LDA. The reactiveness of public policies across the policy sectors (see Table 1) was done through the lens of nudge theory. The extracted topic models (TM) by an unsupervised machine learning method called Latent Dirichlet Allocation (LDA) aided in gaining deeper insights into the nudges made by various policymaking bodies (illustrated through Tables 3 to 9). Besides, we have analysed the high-frequency words (see Fig 3) to have a better bird's eye view of the public policy focus points in the wake of COVID-19 in India.

High probability (β) words across 14 policy sectors (see Table 1) illustrated the heuristics of policymaking in containing the virus spread. The extraction of heuristics revealed that commonalities in policy nudges were on enforcing lockdown rules, improving surveillance and encouraging the public to wear masks and wash hands frequently. Sector-specific heuristic focussed on maintaining equilibrium within the sector. For example, in the agriculture sector,

a critical nudge on allowing the harvest of winter crops for food security amidst lockdown (see Table 3, Agriculture and Food, Topic 1). Heuristics were also extracted in the traditional medicine and well-being sector, that nudged people with #YogaAtHome and Ayurveda for immunity boosting (see Fig 4). These nudges were also towards promoting a healthier lifestyle through traditional medicines and practices, that will be important even in post-COVID scenarios.

The public policy nudges in the chemical sector were on ensuring drug surplus, whereas more nudges were given to the industry to fulfil the shortage of medical devices and ventilators. Preservation of the medical supply chain was a critical heuristic. However, the coronavirus pandemic further created a demand for an efficient supply chain of personal protective equipment (PPE), sanitiser and masks (see Table 3, Chemicals). In doing so, new heuristics were added by nudging rural micro, small and medium enterprises (MSMEs) to join the fight against coronavirus by mass-producing PPE and masks. It had critical social justice implications, especially in rural areas where women-led self-help groups are the primary workforce in such MSMEs (see Table 3, social justice). Nudges on the use of AYUSH-based herbal and traditional products also catered to this rural SME ecosystem which is critical for the survival of the economy in the pandemic.

Besides, the populist Prime Minister (PM) frequently nudged the nation on staying at home, adhering to lockdown rules, improving immunity through yoga and Ayurveda and contributing to the PM-CARES fund (see Table 9). A herd effect was created through such nudges where public participation and micro-donations led the fight against COVID-19. Similar nudges for micro-donations through herd effect was also seen in other critical sectors like the manufacturing, commerce, power, construction and pharma.

Topic extractions also showed herd effect-based policies in the education sector, especially with a higher emphasis on online learning and #StayHomeWithBooks initiatives by the Ministry of Human Resource Development (see Table 3, MHRD). Public broadcasters began to air 80s epic Hindu-epic for herd effect on staying at home with family. Nudges through 'nostalgia' was a significant reactive policy step by the Ministry of Information and Broadcasting (see Table 3, Electronics & IT) to motivate self-isolation. Reactive policies were also seen in the urban sector that nudged municipal authorities to leverage smart technologies like drones for disinfection and surveillance, GIS-platforms and contact tracing apps (see Table 4 and Fig 6).

A herd-effect was also created in the science and technology (S&T) community of India through funding R&D of diagnostic kits, disinfectant coating, crowdsourcing ideas and innovation challenges (see Table 7). Health sector policies focused on aggressive nudging the public to wear homemade masks, maintain social distancing and adhere to hand hygiene rules (see Table 8). The herd-effect was on sensitising people on the severity of COVID-19 transmission for 1.3 billion people.

The Indian Railways acted as a lifeline in ensuring the resilience of the supply chain of essential goods and rapid infrastructure development by converting old trains into isolation wards (see Fig 10 and Table 6). Similarly, the Ministry of Defence and Ministry of Civil Aviation showed reactive policies through joint operations on-air delivery of essential medicine and devices through 'Lifeline UDAAN' mission (see Table 6). It created a herd effect on food and medicine security amongst the public that in turn prevented from hoarding on to essential goods. A critical heuristic in ensuring public follows the national lockdown norms that enabled the efforts of Ministry of Home Affairs (see Table 4).

Our LDA application identifies the herd-effects and policy nudges that can aid in lockdown easement planning, as aforementioned. Similar nudge-based policy approach is especially crucial in a democracy in India with a vast demographic and geo-spatial divide.

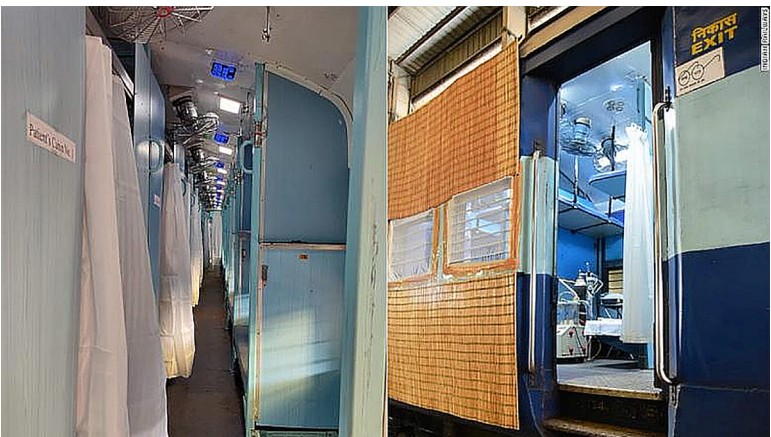

**Fig 10. Indian Railways converted old trains into isolation wards.** (source: [*45*]).

## Conclusion

This study showed an application of topic modelling for public policy. Our application of LDA on government press releases extracted topics across core policy sectors in India that acted as critical nudges in the wake of coronavirus. Use of LDA in such media-data based policy analysis showed its strength in extracting topics that have high concordance with the broader narrative of the government. Our analysis showed that these narratives and nudges created herd effects that motivated the nation of 1.3 billion people to stay home during the national lockdown, even with high economic and social costs.

The integration of computational social science tools like the LDA for identifying nudges for channelizing public behaviour through reactiveness of public policy in the wake of coronavirus outbreak expands the scope of machine learning and AI for public policy applications. From a behavioural public policy perspective, the stochastic interpretation of the topic models through LDA derived critical policy heuristics that must be leveraged during the lockdown easement planning. We believe we are the first in applying LDA to account the reactiveness of COVID-19 induced public policy at multi-sectoral scale. The key conclusions that can be drawn from this study are:

- The use of rigorous media campaigns primarily generated the herd behaviour for successful containment of COVID-19, frequent reminders through SMS, publicising data-driven risk maps generated from innovation grants, public reassurances by the medical community and invoking the feeling of nationalism and solidarity.

- Most of the interventions were targeted to generate endogenous nudges by using external triggers which potentially produces lasting desired behaviour in repeat settings (i.e. repeated broadcasting of information through multi-media channel) and hence can be applied in toto for future challenges.

- Prime Minister's frequent public appearances and assurances nudged in creating the herd effect across pharma, economic, health and public safety sectors that enabled strict national lockdown. It created a herd effect of public participation and micro-donations to the PM-CARES fund to fight the pandemic.

- Successful herd effect nudging was observed around the public health sector (e.g., compulsory wearing of masks in public spaces; Yoga and Ayurveda for boosting immunity),

transport sector (e.g., old railway coaches converted to isolation wards), micro, small and medium enterprises (e.g., rapid production of PPE and masks for frontline words), science and technology sector (e.g., the rapid development of indigenous diagnostic kits, use of robots and nano-technology to fight infection), home affairs (e.g., people adhering to strict lockdown rules even at high economic distress), urban (e.g., drones, GIS-mapping, crowd-sourcing) and education (e.g., work from home and online learning).

- Similar nudging-based approach to the public policy during lockdown easement planning can aid in the smooth yet staggered transition to normalcy. It can even provide a way forward for reviving the economy and climate change mitigation goals in post-COVID era.

- LDA can extract topics that have high concordance to nudges making it a suitable tool to study reactiveness of behavioural public policies.

While this study showed the application of topic models in reactive public policy analysis, the inherent limitations of unsupervised topic modelling remain in the analysis. It interprets the topic models sensitive to the viewpoint of the analysts. Besides, the official press releases used in this study as the primary dataset may contain confirmatory biases, removal of such biases was beyond the scope of this study. The media releases in the Press Information Bureau platform lacked granularity as they are intended for informing the public and media. Another limitation lies in the interpretivist scope of this study when dealing with policy nudges. Nudges are characteristically subjective, and their objective-oriented treatment through our data-driven route may have missed deeper nuances. Such nuances can be efficiently identified by an experienced qualitative researcher. However, it can become manually intensive and unverifiable for a big data corpus.

We also acknowledge that a pure data-driven approach to understanding behavioural attributes like nudges from a big data text corpus can under-represent the problem due epistemological correlations associated with policy documents. Such correlations can induce encoding and ontological biases. For example, epistemic attachment to the object of research can also misinterpret the derived topic models. It will further affect the extraction of critical nudges. Future work is needed in addressing such sensitivity issues in textual data-driven policy analysis.

Nonetheless, this study provided a robust account of the multi-dimensional policy stakes at a national level, especially for a populous and vast country like India. The findings of this paper could be useful for the countries which are in the first stage of this pandemic. Also critical for building resilience framework for future national emergencies from climate change and disasters.

## Author Contributions

**Conceptualization:** Ramit Debnath, Ronita Bardhan.

**Data curation:** Ramit Debnath.

**Formal analysis:** Ramit Debnath.

**Funding acquisition:** Ramit Debnath.

**Investigation:** Ramit Debnath.

**Methodology:** Ramit Debnath, Ronita Bardhan.

**Project administration:** Ramit Debnath, Ronita Bardhan.

**Resources:** Ramit Debnath.

**Software:** Ramit Debnath.

**Visualization:** Ramit Debnath.

**Writing – original draft:** Ramit Debnath.

**Writing – review & editing:** Ronita Bardhan.

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
