## [Decision Letter · Decision Letter 0]

24 Jul 2020

PONE-D-20-13777

India nudges to contain COVID-19 pandemic: a reactive public policy analysis using machine-learning based topic modelling

PLOS ONE

Dear Dr. Bardhan,

Thank you for submitting your manuscript to PLOS ONE. After careful consideration, we feel that it has merit but does not fully meet PLOS ONE’s publication criteria as it currently stands. Therefore, we invite you to submit a revised version of the manuscript that addresses the points raised during the review process.

We look forward to receiving your revised manuscript.

Kind regards,

William Joe

Academic Editor

PLOS ONE

Journal Requirements:

Additional Editor Comments (if provided

2. Please ensure that you refer to Figure 2 in your text as, if accepted, production will need this reference to link the reader to the figure.

3. Please include a copy of Table 10 which you refer to in your text on page 27.

4. We note you have included a table to which you do not refer in the text of your manuscript. Please ensure that you refer to Table 2 in your text; if accepted, production will need this reference to link the reader to the Table.

5. Please note that in order to use the direct billing option the corresponding author must be affiliated with the chosen institute. Please either amend your manuscript or remove this option (via Edit Submission).

Reviewers' comments:

Reviewer's Responses to Questions

**Comments to the Author**

1. Is the manuscript technically sound, and do the data support the conclusions?

Reviewer #1: Yes

Reviewer #2: Yes

2. Has the statistical analysis been performed appropriately and rigorously? 

Reviewer #1: Yes

Reviewer #2: Yes

3. Have the authors made all data underlying the findings in their manuscript fully available?

Reviewer #1: Yes

Reviewer #2: No

4. Is the manuscript presented in an intelligible fashion and written in standard English?

Reviewer #1: No

Reviewer #2: Yes

5. Review Comments to the Author

Reviewer #1: Please refer to the attached annotated manuscript for my feedback.

Reviewer #2: COVID-19 did pose a challenge to the policymakers. Government of India proactively responded to the pandemic. The policy response was heavily based on nudge theory which uses positive and negative reinforcements to modify the behaviour of the people. The authors have done a commendable job of highlighting the manner in which the government was able to control the spread of the virus in India. The findings of this paper could be useful for the countries which are in the first stage. The work merits publication. However, the authors need to highlight the short comings of the methods more clearly.

6. PLOS authors have the option to publish the peer review history of their article (what does this mean?). If published, this will include your full peer review and any attached files.

Reviewer #1: No

Reviewer #2: No

---

## [Author Response · Author response to Decision Letter 0]

25 Jul 2020

Editor

1 Please ensure that your manuscript meets PLOS ONE's style requirements, including those for file naming. The PLOS ONE style templates can be found at 

R: Thank you for this suggestion. We have now formatted the paper according to it. 

2. Please ensure that you refer to Figure 2 in your text as, if accepted, production will need this reference to link the reader to the figure

R: Thank you so much for this comment. We have now carefully checked for such cross-reference errors and amended them. 

3. Please include a copy of Table 10 which you refer to in your text on page 27.

R: Thank you so much for pointing it out. We apologies for a misplaced reference, which have been corrected now. 

4. We note you have included a table to which you do not refer in the text of your manuscript. Please ensure that you refer to Table 2 in your text; if accepted, production will need this reference to link the reader to the Table.

R: Thank you for this comment. We have now corrected it. 

5. Please note that in order to use the direct billing option the corresponding author must be affiliated with the chosen institute. Please either amend your manuscript or remove this option (via Edit Submission).

R: Thank you, we have amended it. 

Reviewer 1

1. This study appears to have already been published by these authors in the journal Computers and Society (see link below). Am I not understanding something?

https //arxiv.org/abs/2005.06619

R: We thank the reviewer for this comment. The attached link is that of the pre-print version of this paper uploaded to the ‘Computers and Society’ section of the arxiv.org server. It is not a journal. 

2. Consider adding the date that these remarks were made to add more context.

R: Thank you for this valuable comment. As suggested, we have now added the date of these lines as,

‘…The Prime Minister of the country, Mr Narendra Modi, in his address to the nation on 24th March 2020, appealed to the nation…’ (see page 2 of the revised manuscript)

3. Revise "population" to "people".

R: Thank you for this correction. We have now made the changes. 

4. Consider revising these sentences for clarity.

R: We are grateful to the reviewer for pointing this out. Revising these sentences have improved the clarity and readability of this paragraph. The revised sentences are as follows: 

“…Nudging is a design-based public policy approach which uses positive and negative reinforcements to modify the behaviour of the people. This approach has a high degree of subjectivity which makes it challenging to ascertain its reliability and replicability under public emergencies like pandemic, disaster, public unrest, etcetera (3). Therefore, it is important to objectively untangle the nudges produced by government policies for efficiently handling national challenges like the COVID-19 pandemic…” (see page 2 in the revised manuscript). 

5. Consider revising this sentence for structure as "big data" is not the technique

R: Thank you for this clever observation. We have now removed the term ‘big data’ to improve the readability of the sentence. 

6. Respectfully " ..harvest intricate nudges..." does not make a lot of sense.

R: We thank the reviewer for pointing it out. We have now amended the sentence for clarity. It is can be referred in page 2- line 55of the revised manuscript as, 

‘…Machine learning (ML) have proven to be a reliable technique in mining and distilling patterns in data and transform into predictive analytics for evidence-based policymaking. This technique is now widely used in deriving crucial information from big data into meaningful policy metrics. We have applied it to extract crucial nudges from official policy response and media releases of the GoI through its nodal agency - Press Information Bureau of India (PIB) (4). This application of ML-based technique for nudge identification from government press releases defines the novelty of this study. The specific ML-technique employed in this study is called topic modelling (TM)…”

7. This section describes methods and would be better placed in that section

R: We have made the required amendments. Thank you for this comment, removing this section has significantly improved the readability of the section 1. 

8. Would note that topic modeling need not be unsupervised necessarily. There are supervised and semi-supervised applications.

R: Thank you for this comment and clearing the associated ambiguity. We have amended the sentence as 

‘…TM is a computational social science method that has its basis in text mining and natural language processing. It automatically analyses text data to determine cluster words for a set of documents (7) …’ (see page 2, line 61)

9. Define as abbreviation at the first use.

R: Thank you for it, we have made the appropriate amendments throughout the text.

10. The point of this sentence is unclear.

R: We are grateful to the reviewer for this comment. We have now removed these sentences and amended the last paragraph of section 1. Thank you so much for pointing it out. 

11. These statements in the introduction section are premature and would be more appropriate for the discussion section.

R: Thank you for this comment, we have now made the amendments and added them to the discussion section. 

12. Would it not have been useful to include the terms "corona virus" and "coronavirus"?

R: We thank the reviewer for this comment and a very important suggestion. The term ‘coronavirus’ was already included in the submitted version of the manuscript. Kindly, see page 3, line 83.

13. The foundation of this analysis seems to be somewhat predicated on the mutual exclusivity of these policy categories. Is it a reasonable assumption that these categories have no overlap?

R: Thank you for this important comment. We are grateful to your observations that have significantly improved our manuscript. Now answering your question, 

Yes, we find the assumption reasonable from the interpretivist nature of our analysis. Most policies are made by different ministries which are structurally and institutionally independent entities. However, we do acknowledge the fact that there will be incidental overlaps as the common goal during this period was to tackle COVID-19 in the country. 

14. Are topics here being equated to the policy categories from Table 1?

R: No, we did not take that route. Instead, we dived deeper into the policy categories to extract the topics that can best illustrate the nudges within that category.

15. I have a hard time believing that the topics are uncorrelated in actuality. This would seem to violate the conditions for using LDA. Can the authors explain either why this was not the case or how the application of LDA required assumptions.

R: Thank you so much for this comment. We read the section again, and indeed it was creating confusion. We are grateful that you could pin-point this ambiguity and guided us in improving the flow of this section. 

We have now amended it as, 

“…The objective of TM is to extract latent semantic topics from large volumes of textual documents (i.e., corpora). LDA is a widely used unsupervised TM technique, with recent applications spanning across political science and rhetoric analysis (8–10,15,16), disaster management (12,17,18) and public policy (13,14,19). It is a generative probabilistic method for modelling a corpus that assigns topics to documents and generates distributions over words given a collection of texts….” (line 111- 115, page 5)

16. Please describe the extent and role of the authors' judgment here.

R: Thank you so much for this suggestion. We have now added a brief explanation to it as, 

“…We used our judgement to coarse estimate the total number of topics under each policy categories through a manually iterative process of reading the policy briefings. Following which the ldatuning (v0.2.0) package (20) in R (v3.5.3) was used to determine the number of topics in each of the topic models…” (line 123-125, page 5)

17. Here and throughout please be sure to cite the version of R and the version of the function.

R: We thank you for this suggestion. We have now updated with the version of the R and the functions. 

18. Unless this is referring to a point in the present document this should be cited with the source document and location of the relevant text.

R: Thank you so much for this comment. We have made the corrections. 

19. This does not seem to have direct bearing on the methods and I am not sure why it is here.

R: We are grateful to the reviewer for such detailed comments and guiding us in the right direction. We have amended this section as per the valuable suggestions. 

20. None of this highlighted text belongs in the methods section.

R: Following the above response, we acknowledge this misplacement and made the necessary corrections. We are extremely grateful to the reviewer for pointing it out. 

21. Is "corpora" misplaced here?

R: Thank you for it, we have corrected this mistake. 

22. On what basis is less than or equal to 50 instances of a word considered "high frequency"? Is this just an arbitrary threshold. If so then perhaps a threshold based on the number of total words or based on the number of words per topic would be more appropriate

R: Thank you so much for this comment. Indeed, we missed an crucial reasoning here, we have now added an explanation as,

“…This threshold was decided based on the total number of unique words and the mode of its repetition in the text corpus…” (line 174 page 7)

23. This has already been defined as PIB earlier in the report.

R: Thank you for this observation, we have now amended it. 

24. What does this mean to have "modelled" the content?

R: Thank for pointing this ambiguity out. We have now amended the sentence in line 205 page 10. 

Reviewer 2

1. COVID-19 did pose a challenge to the policymakers. Government of India proactively responded to the pandemic. The policy response was heavily based on nudge theory which uses positive and negative reinforcements to modify the behaviour of the people. The authors have done a commendable job of highlighting the manner in which the government was able to control the spread of the virus in India. The findings of this paper could be useful for the countries which are in the first stage. The work merits publication. However, the authors need to highlight the short comings of the methods more clearly.

R: We are grateful to the reviewer for supporting our paper and highlighting the merits. We are humbled by your appreciation for our work. Thank you for the comments, we have now further elaborated the limitation section of this method and amended the sections accordingly. 

“…While this study showed the application of topic models in reactive public policy analysis, the inherent limitations of unsupervised topic modelling remain in the analysis. It interprets the topic models sensitive to the viewpoint of the analysts. Besides, the official press releases used in this study as the primary dataset may contain confirmatory biases, removal of such biases was beyond the scope of this study. The media releases in the Press Information Bureau platform lacked granularity as they are intended for informing the public and media. Another limitation lies in the interpretivist scope of this study when dealing with policy nudges. Nudges are characteristically subjective, and their objective-oriented treatment through our data-driven route may have missed deeper nuances. Such nuances can be efficiently identified by an experienced qualitative researcher. However, it can become manually intensive and unverifiable for a big data corpus. 

We also acknowledge that a pure data-driven approach to understanding behavioural attributes like nudges from a big data text corpus can under-represent the problem due epistemological correlations associated with policy documents. Such correlations can induce encoding and ontological biases. For example, epistemic attachment to the object of research can also misinterpret the derived topic models. It will further affect the extraction of critical nudges. Future work is needed in addressing such sensitivity issues in textual data-driven policy analysis. 

Nonetheless, this study provided a robust account of the multi-dimensional policy stakes at a national level, especially for a populous and vast country like India. The findings of this paper could be useful for the countries which are in the first stage of this pandemic. Also critical for building resilience framework for future national emergencies from climate change and disasters…”

---

## [Decision Letter · Decision Letter 1]

28 Aug 2020

India nudges to contain COVID-19 pandemic: a reactive public policy analysis using machine-learning based topic modelling

PONE-D-20-13777R1

Dear Dr. Bardhan,

We’re pleased to inform you that your manuscript has been judged scientifically suitable for publication and will be formally accepted for publication once it meets all outstanding technical requirements.

Kind regards,

William Joe

Academic Editor

PLOS ONE

Additional Editor Comments (optional):

A colored heat map may be provided, if available.

Reviewers' comments:

Reviewer's Responses to Questions

**Comments to the Author**

1. If the authors have adequately addressed your comments raised in a previous round of review and you feel that this manuscript is now acceptable for publication, you may indicate that here to bypass the “Comments to the Author” section, enter your conflict of interest statement in the “Confidential to Editor” section, and submit your "Accept" recommendation.

Reviewer #1: All comments have been addressed

Reviewer #2: All comments have been addressed

2. Is the manuscript technically sound, and do the data support the conclusions?

Reviewer #1: (No Response)

Reviewer #2: (No Response)

3. Has the statistical analysis been performed appropriately and rigorously? 

Reviewer #1: (No Response)

Reviewer #2: (No Response)

4. Have the authors made all data underlying the findings in their manuscript fully available?

Reviewer #1: (No Response)

Reviewer #2: (No Response)

5. Is the manuscript presented in an intelligible fashion and written in standard English?

Reviewer #1: (No Response)

Reviewer #2: (No Response)

6. Review Comments to the Author

Reviewer #1: (No Response)

Reviewer #2: (No Response)

7. PLOS authors have the option to publish the peer review history of their article (what does this mean?). If published, this will include your full peer review and any attached files.

Reviewer #1: No

Reviewer #2: No

---

## [Editor Report · Acceptance letter]

1 Sep 2020

PONE-D-20-13777R1 

India nudges to contain COVID-19 pandemic: a reactive public policy analysis using machine-learning based topic modelling 

Dear Dr. Bardhan:

I'm pleased to inform you that your manuscript has been deemed suitable for publication in PLOS ONE. Congratulations! Your manuscript is now with our production department. 

Kind regards, 

on behalf of

Dr. William Joe 

Academic Editor

PLOS ONE